# Africanized honeybee population (*Apis mellifera* L.) in Nicaragua: Forewing length and mitotype lineages

**Christiane Düttmann[1], Byron Flores [1]\*, Jessica Sheleby-Elías[1], Gladys Castillo[1], Daymara Rodriguez[2], Matías Maggi[3,4], Jorge Demedio[2]**

**1** Centro Veterinario de Diagnóstico e Investigación (CEVEDI), Escuela de Ciencias Agrarias y Veterinarias, Universidad Nacional Autónoma de Nicaragua-León, León, Nicaragua, **2** Facultad de Medicina Veterinaria, Universidad Agraria de La Habana, La Habana, Cuba, **3** Instituto de Investigaciones en Producción Sanidad y Ambiente (IIPROSAM CONICET-UNMdP); Facultad de Ciencias Exactas y Naturales–UNMdP; Centro Científico Tecnológico Mar del Plata–CONICET; Centro de Asociación Simple CIC PBA, Mar del Plata, Argentina, **4** Centro de Investigaciones en Abejas Sociales, Facultad de Ciencias Exactas y Naturales, Universidad Nacional de Mar del Plata, Mar del Plata, Argentina

\* byronfloressomarriba@gmail.com

**Data Availability Statement:** All relevant data are within the manuscript and its Supporting Information files.

## Abstract

Various subspecies of *Apis mellifera* L. were introduced to Central America since colonization 500 years ago. Hybridization increased with the entrance of the Africanized bee in Nicaragua in 1984. Rustic beekeeping activities and numerous feral swarms define the genetic pattern, reflected in phenotypic heterogeneity and remarkable differences in the behaviour of the bee colonies, especially the nest defence. Due to these facts, the question emerge about the degree of Africanization of honeybee colonies in Nicaragua. In this study, we identified Africanized honeybee colonies based on the single character "mean forewing length" and we corroborated our results by determining mitotypes using mtDNA analysis. Morphometric and genetic approaches were realized in three different geographical zones of Nicaragua and related to beehive characteristics and management. Worker bee samples were taken from the inside of 146 hives from 26 apiaries. Abdominal colour as phenotypic character was the first examination, followed by measurement of 1460 right forewings to determine corresponding probability of Africanization. More than 60% of the beehives showed phenotypic heterogeneity and mean forewing length of 8.74 mm (SD 0.16 mm) indicated a high degree of Africanization. Those results provided a selection of 96 worker bees to perform PCR of two worker bees per hive. For mitochondrial DNA analysis 14 samples from sentinel apiaries were added. Three from 61 beehives presented bees with different mtDNA. Throughout, three mitotypes of the African (A) lineage were detected; one mitotype is still unidentified. Mitotype A1 *A. mellifera iberiensis* was represented by 88 bees and mitotype A4 *A. mellifera scutellata* by 21 bees. Phylogenetic analysis confirmed the PCR findings. No associations were found between mitotypes, forewing length, beehive characteristics and management. A high degree of Africanization in *A. mellifera* colonies represented by two predominating mitotypes from the A lineage, prevail in Neotropical Nicaragua, with mitotype A4 predominating at higher altitudes.

**Funding:** The author(s) received no specific funding for this work.

**Competing interests:** The authors have declared that no competing interests exist.

## Introduction

The native range of the Western honeybee (*Apis mellifera*) is broad, comprising Africa, most of Europe and the Middle East. However, the evolutionary origin of the subspecies is still debated, whether it was in Middle East [1–4] or in Africa [5, 6] or even Middle East/North-eastern Africa [7].

Studies based on morphometric [3] and genetic analyses [8, 9], identified five evolutionary lineages of *A. mellifera* (hereinafter referred to as lineage A, M, C. O and Y). The identification of existing subspecies of the Western honeybee over the past 30 years has led to diverse results, due to the use of different identification approaches. With the Morphometric Bee Data Bank in Oberursel, Germany, Ruttner has created a solid framework for the classification of the subspecies of *A. mellifera* [3]. Up to 40 distinct morphometric characters, manually measured, were correlated to geographic variability for honeybee subspecies. Technical progress made computer assisted measurement and analysis possible [10–12], and ongoing morphometric research about *A. mellifera* species focused on geometric morphometric techniques using landmark-based features for subspecies identification [13, 14]. The combination of feature selection techniques (relevant morphometric characters for classification) and classifiers (machine learning algorithm) is used to improve the honeybee subspecies identification [15].

Genetic research offered new possibilities to infer phylogenetic relationships between honeybee species; in particular, mitochondrial DNA (mtDNA) analysis is effective due to maternal inheritance in bee colonies [1]. The intergenic COI-COII region of the mtDNA of *A. mellifera* is preferably investigated due to the variable number of a 192–196 bp sequence (Q) and the complete or partial deletion of the 67 bp sequence (Po). Length variability combined with a restriction site polymorphism enables a quick and easy test to characterize mtDNA mitotypes. Beside mtDNA analysis, nuclear markers are used in population genetic studies. The analysis of the microsatellites (simple sequence repeats, SSR) is able to identify the variation in the number and type of repetitions of the loci and thus can determine the genetic diversity in honeybee populations [8]. Single Nucleotide Polymorphism (SNP) are molecular genetic markers used for genetic diversity research. SNP analysis in bees, identify the sites where they differ in a single nucleotide in nuclear DNA sequence, leading to different genotypes [4]. In countries with lower budget for investigation, an alternative is the sequencing of the genetic code of the mtDNA intergenic COI-COII region [16]. In addition to the studies focused on one approach, investigations with morphometric and genetic approaches were conducted in order to identify subspecies of honeybees, to compare populations at different times and in different regions and to determine the process of Africanization [17–19].

Currently, the number and geographical distribution of honeybee subspecies is not clear due to population dynamics in transition zones [20] and the human impact on genetic diversity due to commercial activities [21], nonetheless the present taxonomic classification of *A. mellifera* mainly describes ~ 30 subspecies [5, 21].

In America, the evolutionary process did not take place as it did in Europe and Africa, because *A. mellifera* is not originated there. Honeybees came to America with European colonists and have spread across the continent for the past 500 years. The European subspecies brought to America were likely *A. mellifera mellifera* (France, Germany) and *A. mellifera iberica* (actually *A. m iberiensis*) and later the Italian bees, *A. mellifera ligustica* [22].

If the introduction of the Western honeybee to the American continent by the European colonialists was a successful biological invasion, the later introduction of the African subspecies into Brazil became an even more decisive event. In 1956, tropical African queens of the subspecies *A. mellifera scutellata* were introduced to exchange genetics with resident European stocks of honeybees to improve honey production [23]. By crossing European honeybee

species with African queens and drones, the Africanized bee emerged and then spread into South, Central and North America [13, 24].

Quality methods measuring a significant number of morphometric characters have discriminate power to identify subspecies, but they are time consuming. Meixner et al. outlined the documented methods for discrimination of morphometric subspecies and lineages of the honeybee by measuring size characters of different body parts, colouration patterns, characteristics of pilosity and a detailed wing shape analysis [19]. In order to differentiate between Africanized and European honeybees as a study purpose, Rinderer et al. developed a simplified method by measuring the forewing length. This single character compared to multivariate analysis showed to be the best criterion to identify Africanization (at p<0.1 level no misidentification) with the overall average of the forewing length of an Africanized bee colony $\bar{x}$ = 8.87 mm and $\bar{x}$ = 9.20 for European bees [25]. Even assuming that this technique provides only preliminary results [19] and that it is unsuitable for identifying subspecies, it does indicate the probability that a colony is Africanized [26].

Genetic investigations are more accurate to discriminate the African and European matriline identities using PCR-RFLP and sequencing analysis. Widely used is the study of the intergenic region of mitochondrial DNA (mtDNA) COI and COII, because of its high degree of genetic variation between the different lineages. The region is composed of a P and/or Q sequence; depending on presence or absence (P-sequence), as well numbers and variation (Q-sequence) the mitotypes within each lineage can be determined [27, 28]. Studies with different restriction enzymes for digestion were conducted during the past 3 decades [18, 19, 29], but the *Dra*I method showed a significant amount of polymorphism detection and it is highly suitable to discriminate Africanized colonies [27, 28].

In Nicaragua, the Africanized honeybee was first reported in 1984 [30]. Due to the high defensiveness of the Africanized bees, in the following years beekeeping activity was reduced over 50% (C. Düttmann personal communication to beekeeper committee), like in many other Latin-American countries where the Africanization of *A. mellifera* had occurred [18]. Since Nicaraguan beekeepers do not practice selective breeding and the majority catch feral swarms to increase the number of their hives, a process of natural selection began for the benefit of the Africanized bee, which replaced the European colonies nationwide. Despite the high level of defensiveness and swarming behaviour, the beekeepers got to know the advantages of working with the Africanized bee, for example resistance and tolerance to diseases, high reproductive numbers and higher honey yields under appropriate management conditions [30, 31].

Although the presence of the Africanized honeybee in Nicaragua is well known, there is still no published information about their morphology and genetic variability. It is important to study the genetic structure of the honeybee populations in Nicaragua, because this information could be used in future genetic improvement programs including the positive traits and attenuating the negative ones of the Africanized honeybee.

The present study wants to give a first insight into the Nicaraguan honeybee population that due to its behaviour characteristics probably has a high Africanized level. The applied methods are measuring of the forewing length, and the determination of mtDNA by PCR and sequencing.

We hypothesize that honeybee population from Nicaragua are mainly composed of Africanized bees, due to rustic apiculture management, where beekeepers capture feral swarms in order to introduce these colonies into their apiaries and furthermore, the practice to exchange frequently combs with brood from strong hives to reinforce weak hives [26, 32, 33].

## Methods

### Study sites

Nicaraguan beekeeping activities are dominated by a tropical savanna climate, with a distinction between three regions characterized by different temperature and precipitation averages and variation in altitudes of the apiary sites. The Pacific Lowlands with a mean temperature of 26.8˚C reaching 45˚C as maximum temperature (tropical savanna climate) and with apiaries at low elevations sites from 20 to 100 meters above sea level (m), exceptionally apiaries on the volcanic chain at 290 m. The Central Zone with lower to similar temperatures (mean 24.9˚C), but tropical monsoon climate during the summer time, and higher elevations than the Pacific Lowlands with apiaries sites from 165 to 375 m. The Northern Highlands with the lowest temperatures (mean 20.4 to 22.8˚C with minimum averages of 14˚C) also tropical monsoon climate and apiaries locations from 280 up to 1009 m [34].

### Sample collection

In order to obtain the specimens for the subspecies identification of *A. mellifera* in three geographically different zones of Nicaragua, samples of worker bees were taken from 146 beehives (26 apiaries): 67 beehives (7 apiaries) in the Pacific Lowlands, 38 beehives (6 apiaries) in the Central Zone and 41 beehives (13 apiaries) in the Northern Highlands. There is more beekeeping activity in the Pacific Lowlands, but the northern region has an extensive variety of microclimates and flora providing different conditions for beekeeping. Samples were collected between 2013 and 2016.

The sample of ~ 250 worker bees per colony was taken from three different frames of the breeding chamber, stored in labeled plastic containers (500 ml) containing 150 ml 96% ethanol and taken to the laboratory of the Centro Veterinario de Diagnóstico e Investigación, (CEVEDI), Escuela de Ciencias Agrarias y Veterinarias (ECAV), Universidad Nacional Autónoma de Nicaragua-León (UNAN-León).

### Data collection about defensive behaviour and beehive management

During the sampling, additional data were registered, such as the defensive behaviour of the colony measured by massive attack during the sampling and persecution after leaving the apiary indicated in meters (modified method, defensive response [35]. The ordinal classification of the defensive behaviour: 1. gentle: no massive attack and low persecution (<30 m) by few bees; 2. moderate: no massive attack, but strong persecution (>30 m) or massive attack, but low persecution (<30 m); 3. very defensive: immediate and high massive attack, followed by strong persecution (>30 m up to more than 100 m). Management information was recorded from the beekeepers about the applied methods of multiplication of the beehives, queen bee replacement and the origin of the queen in the beehive. To collect data, the investigators developed a questionnaire that was pilot–tested before application. The flow chart of investigation procedure is shown in Fig 1.

### Part 1: Phenotypic and morphometric identification

Although it is not used as a main discrimination factor, the abdominal colour variability still is a visual sign for intracolonial diversity, distinguished by the pigmentation variation of the abdominal tergites and the scutellum. The simple observation of the bimodal variation between "yellow" and "dark" honeybees in the same beehive indicates phenotypical diversity. As the first feature in this study, phenotypic diversity of the bee colonies in each of the 146

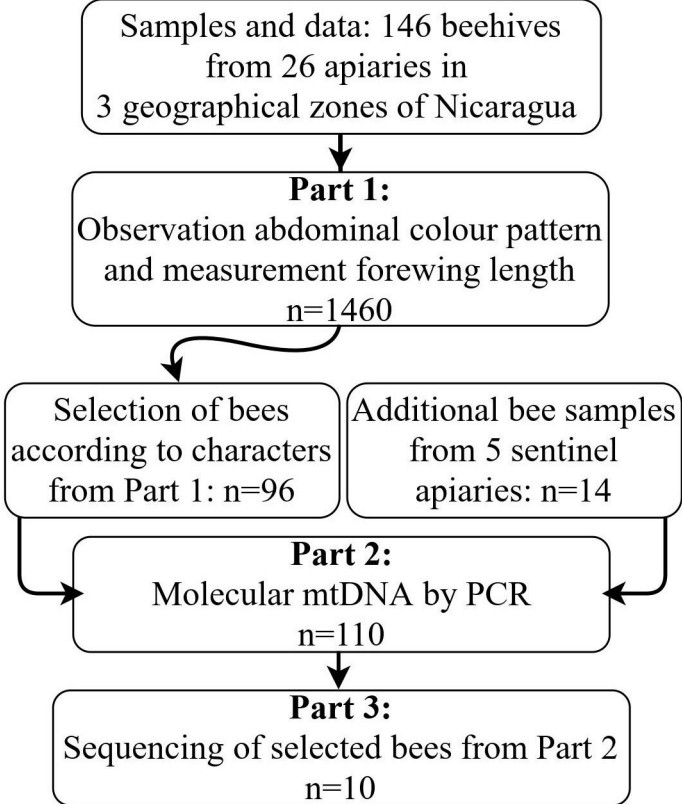

**Fig 1. Flow chart of investigation procedure.**

hives was examined using photographic identification of the collected bees to document uniformity or diversity of the abdominal colour pattern. [3, 19].

As an effective screening tool, the measurement of the length of the right forewing was used as a single character and simple morphometric identification that best discriminate Africanized and European honeybees, as described by Rinderer et al. (1986). From each sample, ten specimens were selected; in the case of phenotypic heterogeneity, half of the bees were dark and the other half yellow. The right forewing of each selected individual was removed as close to the thorax as possible and mounted on a slide, stereoscopically photographed, digitalized and measured with the program Image Tool 3, (S1 Fig), [36].

## Part 2: Molecular mtDNA identification

For molecular diagnosis, honeybees were selected from all inspected apiaries in study. In order to obtain relatively high genetic diversity from the Nicaraguan honeybee population, the selection criteria based on heterogeneity of abdominal colour within the colony and mean length of forewing. Due to the management of beekeepers in Nicaragua, referring to frequently exchanging combs with brood to strengthen weak hives, exists the probability of finding diversity in mitochondrial DNA in one beehive [30]. Therefore, we decided to sample two worker bees per beehive. Phenotypically there is a high diversity within the colonies and in the recollected samples, which might not be only because of polyandry. As Evans et al. described, that in case selected worker bees could be from different colonies, it is worth sampling more than one individual to avoid mistakes in assigning colony heritage [22]. The decision to select more

than one worker bee is a modification of the usual methodology in processing mtDNA with the *Dra*I-Test.

Additionally, only for genetic analyses (but not for forewing length and additional data), honeybees from five sentinel apiaries of the annual monitoring program from the Instituto de Protección y Sanidad Agropecuaria (IPSA) were included, because of the particular location of the apiaries: northern border to Honduras (1), southern border to Costa Rica (1), Island in the Nicaraguan Lake (1) and the South Caribbean Coast (2). Inclusively, the sample size for PCR selection comprised 110 worker bees from 61 beehives from 31 apiaries; and per zone Pacific Lowlands (33), Central Zone (26) and Northern Highlands (51).

**DNA extraction.** The 110 bees preserved in 96% ethanol and stored at −20°C until molecular analysis, were washed three times with PBS and once with distilled water. Total DNA isolation was performed from the two hind legs of each individual following the method of Garnery et al. (1993) and according to the protocol described in the Qiagen Blood and Tissue DNA kits (Qiagen, Germany)) with an adjusted elution volume of 50 μl of each sample instead of 200 μl.

**Mitochondrial DNA analysis.** For PCR amplification of the intergenic COI-COII region E2 forward primer 5′-GGCAGAATAAGTGCATTG-3′ and H2 reverse primer 5′-CAATAT CATTGATGACC-3′ were used. The reaction was carried out in a total volume of 50 μl, and contained 25 μl of Master Mix 2X (Promega, USA), 2 μl each primer (10,000 nM), 16 μL of Nuclease-free water and 5 μl of DNA of the samples. The amplification protocol consisted of an initial temperature of 95°C for 5 minutes, followed by 35 cycles of: 95°C for 30 seconds, 47°C for 30 seconds and 72°C for 1 minute. Subsequently a final elongation at 72°C for 15 minutes and hold at 4°C. The PCR reaction was performed with the Applied Biosystem 2720 Thermocycler. The amplification products were separated by placing 10 μl of the mixture on 1.5% agarose gel with ethidium bromide and visualized under UV light.

The PCR products were digested with *Dra*I restriction enzyme to identify mitotypes of *A. mellifera* at the recognition and cleavage sequence TTTAAA. According to the protocol from Thermo Fisher Scientific (MA, USA), adding 18 μl of nuclease-free water, 2 μl of 10X buffer, 2 μl of the enzyme, 10 μl of the PCR product. It was mixed gently and incubated at 37°C for 16 hours. The digestion products were observed on a 1% agarose gel with ethidium bromide. The determination of the mitotypes was carried out according to the number and size of the DNA segments as described by Garnery et al. (1993).

## Part 3: Sequencing

To confirm the findings in PCR diagnosis, ten specimens were selected for sequencing from the three sample zones: three specimens from the Pacific Lowlands, three from the Central Zone and four from the Northern Highlands. In addition to their geographical origin, the ten specimens were selected based on the results of the determination of the mitotype; likewise, it was considered that in the Central Zone and Northern Highlands the result of the PCR indicated different mitotypes in the same hive.

Reverse-sense sequencing (H2 primer) was performed for the ten selected samples. DNA purifications were carried out with the commercial kit according to the manufacturer Ultra-Clean® 15 DNA Purification (MO BIO, USA). The sequencing was carried out at the Centro de Investigación en Biología Celular y Molecular (CIBCM), Universidad de Costa Rica (UCR).

**Phylogenetic analysis.** The sequences of each sample were aligned using ClustalW 1.6 to study evolutionary history. Evolutionary divergences were calculated using the Maximum Likelihood (ML) method and reported as units of the number of base substitutions per site. Bootstrap test (1000 repetitions) was performed to determine statistical significance and expressed as the percentage of occurrence of each taxon in the branches of the phylogenetic tree.

Evolutionary analysis was conducted in MEGA 7 software [37]. The sequences were sent to the National Center for Biotechnology Information (NCBI) using Banklt (NCBI, 2021), obtaining the identification numbers: MW695396, MW695397, MW695398, MW695399, MW695400, MW695401, MW695402, MW695403, MW695404, and MW695405.

## Statistical analysis

Qualitative variables were expressed in relative frequency. ANOVA test was realized for the mean forewing length related to zones, departments, municipalities of sampling and characteristics of beehives and their management. The degree of Africanization of the beehives was determined according to the table *"Fore-wing length means ($\bar{x}$) of 10 bees and corresponding probabilities of their being Africanized (PA) or* European (EA)" [25]. Pearson correlation test was applied to relate forewing length means with altitude of sampling sites.

Student's t test was applied to compare quantitative variables with normal distribution such as the length of the forewing and the altitude of the locations of the apiaries, both related to the two identified mitotypes.

Chi Square tests ($\chi^2$) were performed to show association between the geographic distribution of the mitotypes and characteristics of the beehives and their management.

A Fisher's exact test was applied to compare the frequency of mitotypes between ten sampled departments of the country, and considered statistically significant at $\alpha = 0.05$. Data management and analyses were performed using R v.3.4.1 statistical software (Foundation for Statistical Computing, Vienna, Austria).

## Ethical approval

The Research Ethics Committee (ECAV, UNAN-León) has previously approved the study. The samples were sourced from 26 private apiaries with the permission of the apiary owners; on the other hand, samples were provided from IPSA from epidemiological monitoring. The animals collected are not considered endangered or protected by any institution.

## Results

The present study reveals a first insight to the Africanization level of honeybee colonies in Nicaragua by measuring the forewing length and identifying the different mitotypes in three zones with beekeeping activities, sustained by findings of the defensive behaviour. The reproduction management of the apiaries is a main activity that contributes to colony diversity, reflected by phenotypic diversity identified by abdominal colour pattern.

### Colony characteristics and beehive management

The defensive behaviour of the studied 146 colonies was related to Africanized characters in *A. mellifera*. Three quarters of the colonies had a defensive tendency; 20.5% were highly defensive with massive attacks during handling the hives and persecutions from 30 up to 400 meters leaving the apiary; and 54.1% of the beehives were moderately defensive, requiring special management to avoid problems. Only 25.3% were gentle bees, without any problem in management.

The obtained data revealed four methods used for multiplication of the 146 beehives. The non-technical method was capturing feral swarms (22.6% of the hives) and 93.9% of them remained without queen replacement. Likewise, beekeepers bought hives from other beekeepers without replacement of the queen bee (17.8% of the hives). The most applied multiplication method was the division of the hives (46.6%); nearly all of the new introduced queens were self-reared by the beekeepers (98.5%).

In 57.5% of the studied hives, the beekeepers replaced the queens and 95.2% of those beekeepers used self-reared queens. Only 4.8% of the queens were purchased from one cooperative with special breeding techniques. There is no queen-rearing program in Nicaragua that could supply selected queens to beekeepers.

The hive characteristics and their management were significantly different in between the sampling zones (Table 1).

Phenotypic diversity was detected in 62.3% of the hives according to the pattern of abdominal colour with variations from dark brown to light yellow in the same colony (Fig 2); there was a higher degree of heterogeneity among the hives in the Pacific Lowlands (Table 1).

## Forewing length

The overall average of the right forewing length ($\bar{x}$ = 8.74 mm; SD = 0.16 mm), measured in 146 sampled beehives, indicated a high degree of Africanization. This study revealed that in Nicaragua more than 79% of the beehives had a degree of 99% of Africanization. The level of 90% of Africanization was detected in 87% of the beehives; there were only 8/146 managed colonies with a European character more than 50%, according to the forewing length (Fig 3). The mean length of the right forewing per beehive was not different among the sample sites (ANOVA, zone: $F_{2,145}$ = 1.77, p > 0.05; department: $F_{6,145}$ = 1.32, p > 0.05; municipality: $F_{13,145}$ = 1.25, p > 0.05) (Table 2), also there was no correlation between forewing length and altitudes of the localization of the apiaries (R = -0.02, p > 0.05). In addition, there was no significant difference between the characteristics of the beehives and management activities according to the forewing length (ANOVA, origin of the colony: $F_{3,146}$ = 0.13, p > 0.05; abdominal colour pattern: $F_{1,146}$ = 0.55, p > 0.05; defensiveness of the beehive: $F_{2,146}$ = 1.52, p > 0.05; queen replacement: $F_{1,146}$ = 0.48, p > 0.05; origin of the queen: $F_{2,146}$ = 0.66, p > 0.05) (Fig 4).

## Mitotype identification

The results obtained by *Dra*I test indicated that 100% of the studied *A. mellifera* population (110 worker bees) had an African mitotype, represented in three classes; 80% (88/110) of the

**Table 1. Association of *Apis mellifera* hive characteristics and management according to sampled zones.**

| Hive characteristics and management | | Sampled zones | | | DF | $\chi^2$ | p |
|---|---|---|---|---|---|---|---|
| | | Pacific | Central | North | | | |
| Abdominal colour pattern | uniform | 4 | 8 | 26 | 2 | 12.39 | < 0.01 |
| | different | 25 | 11 | 22 | | | |
| Defensiveness | gentle | 12 | 0 | 5 | 4 | 29.06 | < 0.01 |
| | moderate | 13 | 17 | 21 | | | |
| | very defensive | 4 | 2 | 22 | | | |
| Origin of the beehive | captured | 14 | 0 | 3 | 6 | 89.32 | < 0.01 |
| | bought | 3 | 9 | 4 | | | |
| | divided | 12 | 1 | 41 | | | |
| | reared | 0 | 9 | 0 | | | |
| Queen replacement | yes | 12 | 10 | 38 | 2 | 12.00 | < 0.01 |
| | no | 17 | 9 | 10 | | | |
| Origin of the queen | feral | 17 | 0 | 6 | 4 | 35.80 | < 0.01 |
| | own | 12 | 17 | 42 | | | |
| | reared | 0 | 2 | 0 | | | |

DF: Degree of freedom, χ2: Statistic value, p: value

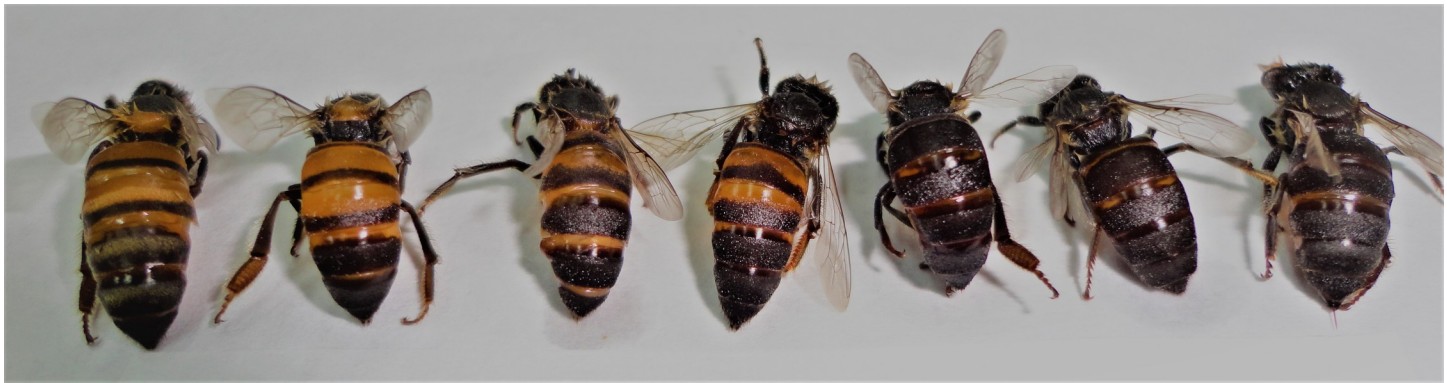

**Fig 2. Worker bees from one beehive (*Apis mellifera*) in Boaco, Central Nicaragua.** Seven worker bees from the same sample of one beehive showing high polymorphism according to abdominal colour pattern.

samples were identified as mitotype A1, while 19.1% (21/110) were identified as mitotype A4. One mitotype is still unidentified and will be submitted to further identification. The distribution of the mitotypes by the sampled zones was significantly different (sampled zones: $\chi^2_{4,110}$ = 10.48, p<0.05). According to $\chi^2$, the mitotype A1 was found with a frequency of 80.8% (21/26) and 97% (32/33) in the Central Zone and Pacific Lowlands respectively, while in the Northern Highlands the frequency of this mitotype was 68.3% (35/51). Concerning the altitude of the apiaries, the distribution of the mitotypes was significantly different (p < 0.001), finding mitotype A4 mostly in higher locations ($\bar{x}$ = 674.78 m, SD = 281.14 m) and the more frequent mitotype A1 in lower zones ($\bar{x}$ = 316.68 m, SD = 275.76 m) (Fig 5).

The association of the mitotypes with the characteristics of the beehives and their management was calculated based on the genetic results of the 96 bees from the main database; 14 mitotypes were excluded, because the information wasn´t complete (honeybee samples with courtesy of epidemiological surveillance by IPSA) (Fig 1). The analyses revealed, that there is no significant difference between the mean forewing length of the identified mitotypes (p > 0.05), mitotype A1 with a mean forewing length $\bar{x}$ = 8.75 mm (SD = 0.18 mm) and mitotype A4 with $\bar{x}$ = 8.79 mm (SD = 0.21 mm). No significant difference was observed linking mitotypes to the defensiveness of the hives and the phenotypic diversity (defensiveness: $\chi^2_{4,96}$ = 3.39 p≥0.05; abdominal colour pattern: $\chi^2_{2,96}$ = 2.68 p≥0.05). Neither did the comparison of the mitotypes with the management of the hives (origin hive: $\chi^2_{6,96}$ = 8.60 p≥0.05; replacement queen: $\chi^2_{2,96}$ = 0.80 p≥0.05; origin queen: $\chi^2_{4,96}$ = 3.11 p≥0.05).

Worker bees with different mtDNA in the same beehive were detected in 3/61 beehives, which belonged to one apiary in the Central Zone and two apiaries in the Northern Highlands.

## Phylogenetic analysis

The phylogenetic tree constructed by the ML method based on inference analysis of the COI-COII intergenic region, classified three bees of the present study as *A. mellifera scutellata* mitotypes A4 (MW695396, MW695397, MW695398) in the same group other sequences were described from Africa (MG592308, FJ477987, KF977009, AF503562), Brazil (EF033650) and Argentina (JQ582439). The sequences of six samples were classified as *A. mellifera iberiensis* mitotypes A1 (MW695399, MW695400, MW695401, MW695402, MW695404, MW695405) clustered with samples from Africa (KX463745, MG592298), Europe (KX463736, FJ477985) and America (EF033649 JQ582438, FJ743639). The

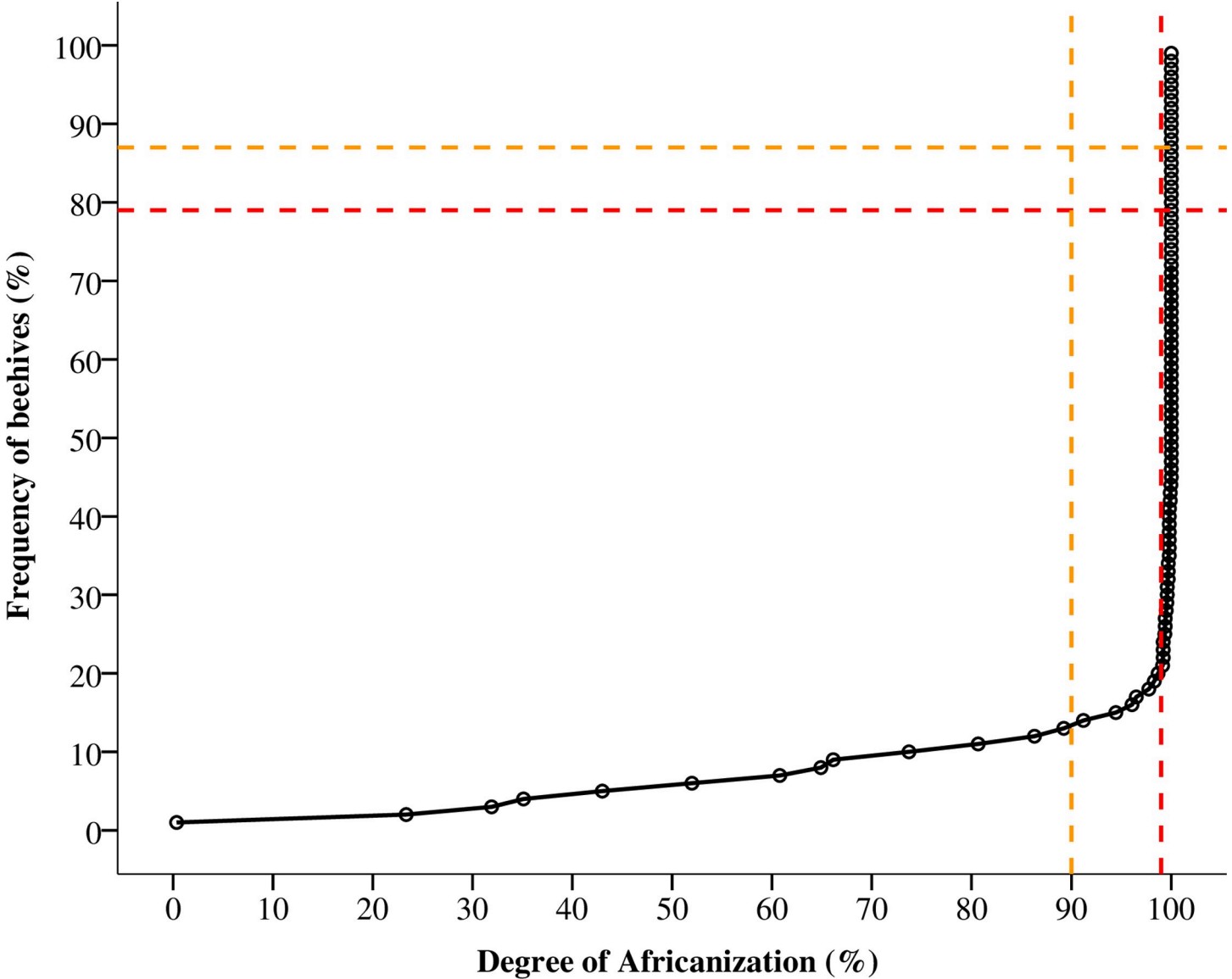

**Fig 3. Degree of Africanization in beehives (*Apis mellifera*) in Nicaragua.** Yellow dotted lines 90% of Africanization; Red dotted lines 99% of Africanization at beehive level.

sequences corresponding to the mitotypes M4 (EF033656), C22 (FJ037785) and C1 (EF033655) were grouped distantly from the Nicaraguan mitotypes A4 and A1 (Fig 6).

The evolutionary history was inferred by using the Maximum Likelihood method and Tamura-Nei model [38]. The tree is drawn to scale, with branch lengths measured in the number of substitutions per site. The proportion of sites, where at least one unambiguous base is present in at least one sequence for each descendent clade, is shown next to each internal node in the tree. This analysis involved 29 nucleotide sequences. Codon positions included were 1st +2nd+3rd+Noncoding. There were a total of 656 positions in the final dataset.

The evolutionary divergence between the three samples from Nicaragua classified as mitotype A4 ranged from 0.000 to 0.003 (similarity more than 99.7%), same result with the sequences from other countries with similarity more than 99.7, exceptionally one Nicaraguan

**Table 2. Beehive mean length of the right forewing from *Apis mellifera* at different location level.**

| Location level | | sampled in | | Mean Wing Length | SD | 95% Confidence Interval | | DF | F | p |
|---|---|---|---|---|---|---|---|---|---|---|
| | | # Apiaries | # Beehives | | | lower bound | upper bound | | | |
| Zone | Pacific | 7 | 67 | 8.74 | 0.16 | 8.70 | 8.78 | 2 | 1.768 | 0.17 |
| | Central | 6 | 38 | 8.71 | 0.16 | 8.66 | 8.77 | | | |
| | North | 13 | 41 | 8.78 | 0.17 | 8.73 | 8.83 | | | |
| Department | León | 4 | 38 | 8.73 | 0.15 | 8.68 | 8.78 | 6 | 1.317 | 0.25 |
| | Chinandega | 3 | 29 | 8.75 | 0.16 | 8.69 | 8.81 | | | |
| | Boaco | 6 | 38 | 8.71 | 0.16 | 8.66 | 8.77 | | | |
| | Matagalpa | 5 | 15 | 8.78 | 0.21 | 8.66 | 8.90 | | | |
| | Jinotega | 5 | 15 | 8.82 | 0.14 | 8.74 | 8.90 | | | |
| | Estelí | 1 | 3 | 8.62 | 0.23 | 8.05 | 9.18 | | | |
| | Madriz | 2 | 8 | 8.77 | 0.07 | 8.71 | 8.83 | | | |
| Municipality | León | 3 | 28 | 8.72 | 0.15 | 8.66 | 8.78 | 13 | 1.250 | 0.25 |
| | Nagarote | 1 | 10 | 8.75 | 0.17 | 8.62 | 8.87 | | | |
| | Villanueva | 3 | 29 | 8.75 | 0.16 | 8.69 | 8.81 | | | |
| | Boaco | 1 | 5 | 8.71 | 0.13 | 8.54 | 8.87 | | | |
| | Teustepe | 5 | 33 | 8.71 | 0.17 | 8.65 | 8.77 | | | |
| | Ciudad Dario | 2 | 6 | 8.66 | 0.05 | 8.61 | 8.71 | | | |
| | Tuma—La Dalia | 1 | 3 | 8.80 | 0.36 | 7.91 | 9.70 | | | |
| | Matiguás | 1 | 3 | 8.81 | 0.23 | 8.23 | 9.40 | | | |
| | Muy Muy | 1 | 3 | 8.94 | 0.21 | 8.42 | 9.47 | | | |
| | Jinotega | 1 | 3 | 8.71 | 0.22 | 8.17 | 9.26 | | | |
| | El Cuá | 3 | 9 | 8.85 | 0.13 | 8.75 | 8.95 | | | |
| | Wiwilí | 1 | 3 | 8.84 | 0.06 | 8.69 | 8.99 | | | |
| | Estelí | 1 | 3 | 8.62 | 0.23 | 8.05 | 9.18 | | | |
| | Somoto | 2 | 8 | 8.77 | 0.07 | 8.71 | 8.83 | | | |

SD: Standard deviation, DF: Degree of freedom, F: Statistic value, p: value significant at <0.05

mitotype A4 (MW695396) that was 100% similar to sequences described in South Africa, Brazil and Argentina (FJ477987, EF033650, JQ582439). The similarity between the six samples from Nicaragua classified as mitotype A1 was at least 99.8%; five of these samples were 100% similar to the A1 mitotype described in Argentina (JQ582438), and four were 100% similar to the sequence from Benin (MG592298). Relating A1 mitotypes from south-western Europe to the Nicaraguan A1 sequences, the divergence was higher (between 0.002 and 0.004), even higher the comparison with the sequence from North Africa (between 0.004 and 0.006). The sample MW695396 (Mitotype A4) and the sample MW695399 (Mitotype A1) were collected from the same hive and presented a divergence of 0.119 (Table 3).

## Discussion

The process of the Africanization of the honeybee *A. mellifera* in the continent of the Americas since the 60´s of the last century began in Nicaragua more than 3 decades ago. The abundant flora of the country and a mostly rustic beekeeping with little intervention from beekeepers, helped honeybees with African characteristics to reproduce easily, especially, when colonies of feral bee populations formed part of the stock managed by beekeepers [18, 26]. Table 1 shows that there are significant differences between the sample zones in terms of defensiveness, the origin of the hive, replacement of the queen and the origin of the same. Curiously, the results

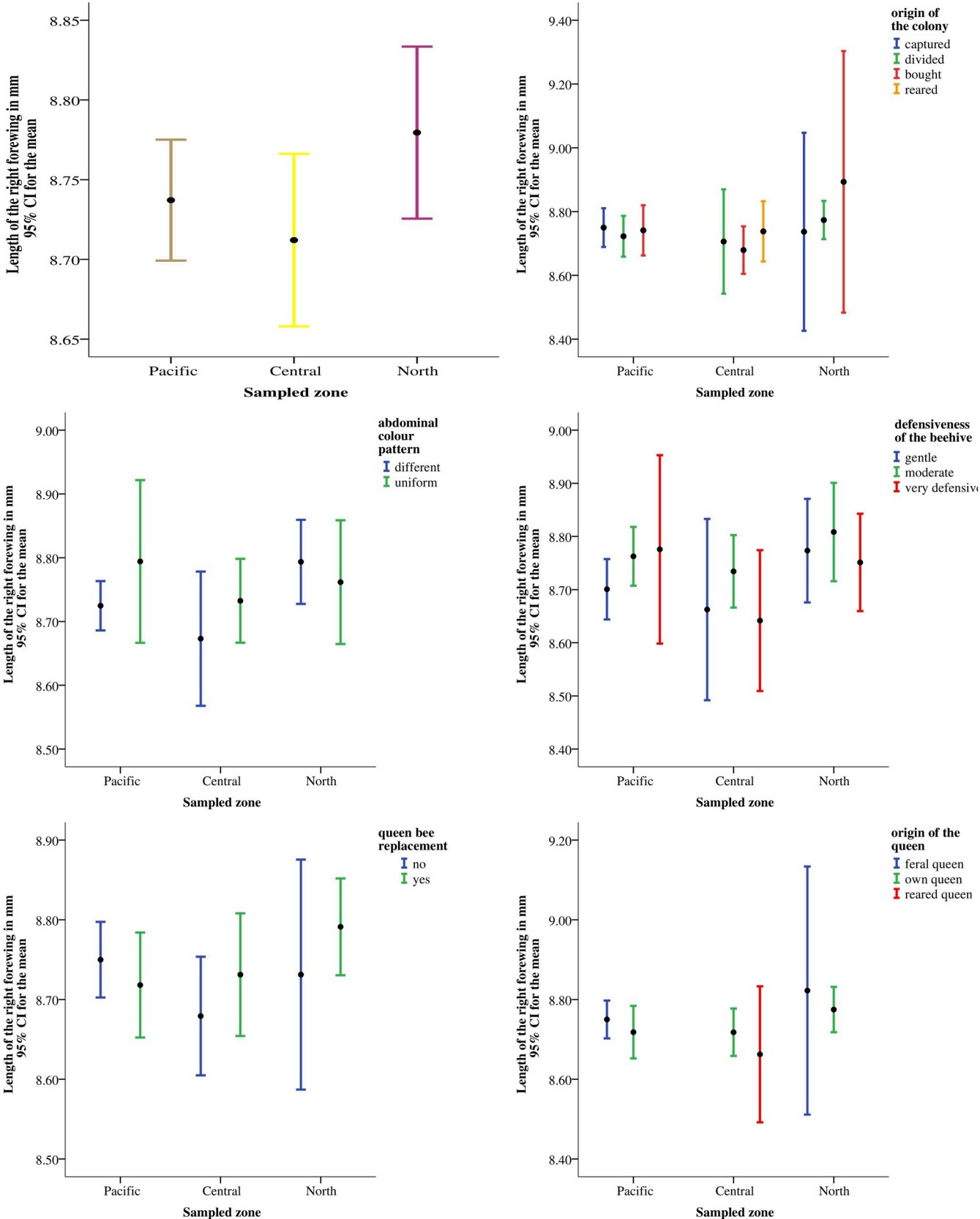

**Fig 4. Beehive characteristics according to sampled zone and related to forewing length.** (a) Forewing length by sampled zone; (b) Forewing length by origin of the colony and sampled zone; (c) Forewing length by abdominal colour pattern and sampled zone; (d) Forewing length by defensiveness of the hive and sampled zone; (e) Forewing length by queen bee replacement and sampled zone; (f) Forewing length by origin of the queen and sampled zone.

reflect that in the Pacific Lowlands bees are docile, although the capture of feral swarms is widely practiced. In contrast, in the Northern Highlands offensive behaviour predominates, although the division of hives with queen replacement is common. It appears that defensive colonies are being reproduced, a finding that may in part be explained by the interest of bee-keepers in avoiding theft of their hives. Queen rearing is merely practiced in the Central Zone of the country; also, there are few beekeepers who buy queens to replace them in new hives. As mentioned before, in the Pacific Lowlands, feral swarms are captured to multiply the hives, this is also reflected in the phenotypic heterogeneity. For this reason, it was decided to select two worker bees per sample for the identification of the mitotypes [27].

Due to these conditions and the results of the present study, in which *A. mellifera* from lineage A coincide with the morphometric character of the mean forewing length $\bar{x}$ = 8.74 mm, a high degree of Africanization can be found in colonies of *A. mellifera* in Nicaragua, confirmed

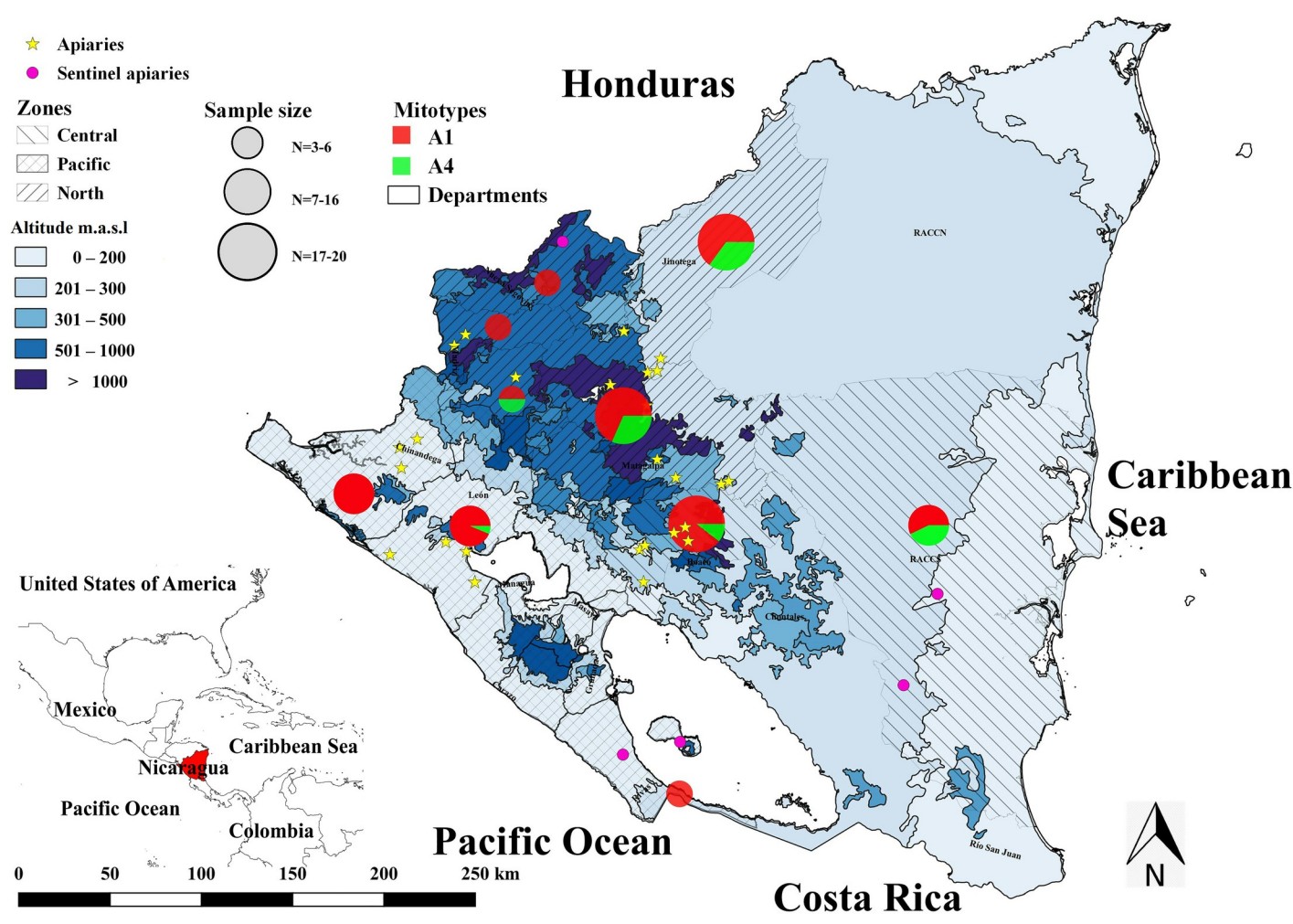

**Fig 5. Geographic distribution of *Apis mellifera* mitotypes by department, Nicaragua.**

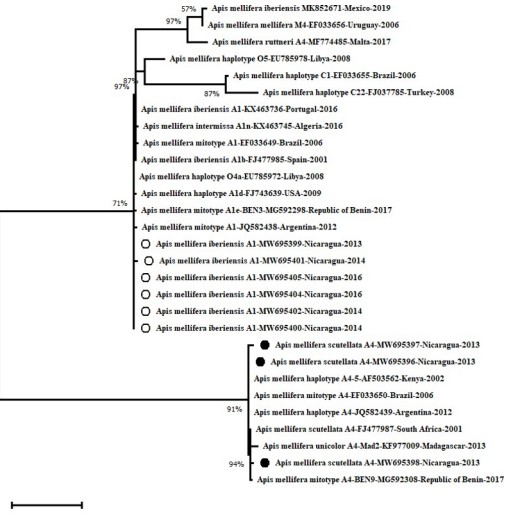

**Fig 6. Phylogenetic analysis of COI-COII gene nucleotide sequences of *Apis mellifera*.**

by the discrimination of Rinderer et al. (1986) who described x̄ = 8.87 mm as the overall average of the forewing length for Africanized beehives.

As Nicaragua has tropical climate, even in higher locations, it is not surprising to find Africanized honeybees across the country. There was no morphological variation nationwide, considering the mean forewing length as an indicator for Africanization [25], since there was no difference between the study areas or the height of the location of the apiaries (Table 2), neither association to characteristics of the hives and nor to their management (Fig 4). Amakpe et al. in Benin, Africa, found no correlation between morphometric characters (mainly leg and wing length) and altitude of the inspected apiaries, although they described a positive south-north gradient in bee size. Moreover, they indicated that there was no difference between the morphometric characters between the different haplotype, same results as in the present study in Nicaragua [39].

This study revealed that all the analysed bees belong to the A lineage, whereas 80% (88/110) of the samples were identified as mitotype A1, and 19.1% (21/110) as mitotype A4. In one sample, mitotype A could not be classified and will be submitted for further identification. There was a significant difference between mitotypes and altitudes corresponding to different zones, more honeybees with mitotype A4 were detected in higher altitudes (x̄ = 674.78 m) of the Northern Highlands than in the Pacific Lowlands and Central Zone (Fig 5). Similarly, Kraus et al. described mitotype A4 more frequently in higher altitudes in Veracruz, Mexico. As a reference of an African population, they cited Franck et al. (2001) reporting 70% of mitotype A4 and 20% A1 at an altitude of 1300 m in Pretoria, South Africa with humid subtropical climate according to Köppen Geiger Climate Classification [40].

In Costa Rica with environmental conditions similar to those of Nicaragua, Lobo found a higher frequency of mitotype A1 (55.9%) than mitotype A4 (33.9%) [41]. In Colombia, 98.3% of the studied bee population had African mitotypes, represented in 17 classes (A), being A1 and A4 mitotype, the most frequent within this lineage [9]. In apiaries from Brazil and Uruguay, Collet et al. reported the predominance of the A1 and A4 mitotypes; however, they found a higher frequency of the A4 mitotype (68%) [42]. This difference can be attributed to the fact that generally A1 mitotype is more common in northern regions, both in Africa and Latin America, while the A4 mitotype has been observed more frequently in South Africa and

**Table 3. Estimation of evolutionary divergence between selected *Apis mellifera* mitotypes.**

| | | 1 | 2 | 3 | 4 | 5 | 6 | 7 | 8 | 9 | 10 | 11 | 12 | 13 | 14 | 15 | 16 | 17 | 18 | 19 |
|---|---|---|---|---|---|---|---|---|---|---|---|---|---|---|---|---|---|---|---|---|
| 1 | A4-MW695397-Nicaragua | | | | | | | | | | | | | | | | | | | |
| 2 | A4-MW695396-Nicaragua | 0.003 | | | | | | | | | | | | | | | | | | |
| 3 | A4-MW695398-Nicaragua | 0.000 | 0.003 | | | | | | | | | | | | | | | | | |
| 4 | A4-MG592308-Benin | 0.002 | 0.002 | 0.002 | | | | | | | | | | | | | | | | |
| 5 | A4-FJ477987-South Africa | 0.002 | 0.000 | 0.002 | 0.000 | | | | | | | | | | | | | | | |
| 6 | A4-EF033650-Brazil | 0.003 | 0.000 | 0.003 | 0.002 | 0.000 | | | | | | | | | | | | | | |
| 7 | A4-JQ582439-Argentina | 0.003 | 0.000 | 0.003 | 0.002 | 0.000 | 0.000 | | | | | | | | | | | | | |
| 8 | A1-MW695405-Nicaragua | 0.118 | 0.108 | 0.118 | 0.112 | 0.108 | 0.108 | 0.108 | | | | | | | | | | | | |
| 9 | A1-MW695404-Nicaragua | 0.117 | 0.107 | 0.117 | 0.111 | 0.107 | 0.107 | 0.107 | 0.000 | | | | | | | | | | | |
| 10 | A1-MW695402-Nicaragua | 0.117 | 0.107 | 0.117 | 0.111 | 0.107 | 0.107 | 0.107 | 0.000 | 0.000 | | | | | | | | | | |
| 11 | A1-MW695401-Nicaragua | 0.116 | 0.110 | 0.116 | 0.114 | 0.111 | 0.110 | 0.110 | 0.002 | 0.002 | 0.002 | | | | | | | | | |
| 12 | A1-MW695400-Nicaragua | 0.117 | 0.107 | 0.117 | 0.111 | 0.107 | 0.107 | 0.107 | 0.000 | 0.000 | 0.000 | 0.002 | | | | | | | | |
| 13 | A1-MW695399-Nicaragua | 0.119 | 0.110 | **0.119** | 0.114 | 0.110 | 0.110 | 0.110 | 0.000 | 0.000 | 0.002 | 0.000 | | | | | | | | |
| 14 | A1-EF033649-Brazil | 0.193 | 0.190 | 0.210 | 0.206 | 0.205 | 0.206 | 0.208 | 0.002 | 0.002 | 0.002 | 0.004 | 0.002 | 0.002 | | | | | | |
| 15 | A1-JQ582438-Argentina | 0.195 | 0.188 | 0.195 | 0.191 | 0.188 | 0.188 | 0.188 | **0.000** | **0.000** | **0.000** | 0.002 | **0.000** | **0.000** | 0.003 | | | | | |
| 16 | A1n-KX463745-Algeria | 0.199 | 0.196 | 0.216 | 0.212 | 0.211 | 0.212 | 0.214 | 0.004 | 0.004 | 0.004 | 0.006 | 0.004 | 0.004 | 0.003 | 0.005 | | | | |
| 17 | A1e-MG592298-Benin | 0.197 | 0.190 | 0.213 | 0.209 | 0.207 | 0.207 | 0.208 | **0.000** | **0.000** | **0.000** | 0.004 | **0.000** | 0.002 | 0.005 | 0.003 | 0.005 | | | |
| 18 | A1b-FJ477985-Spain | 0.168 | 0.164 | 0.168 | 0.163 | 0.162 | 0.163 | 0.164 | 0.002 | 0.002 | 0.002 | 0.004 | 0.002 | 0.002 | 0.000 | 0.002 | 0.002 | 0.003 | | |
| 19 | A1-KX463736-Portugal | 0.197 | 0.194 | 0.214 | 0.210 | 0.209 | 0.210 | 0.212 | 0.002 | 0.002 | 0.002 | 0.004 | 0.002 | 0.002 | 0.002 | 0.003 | 0.002 | 0.003 | 0.000 | |

The number of base differences per site from between sequences are shown. This analysis involved 19 nucleotide sequences. Codon positions included were 1st+2nd +3rd+Noncoding. All ambiguous positions were removed for each sequence pair (pairwise deletion option). There were a total of 656 positions in the final dataset.

South America [8, 42]. Further north from Nicaragua, in Mexico, Dominguez-Ayala et al, studied honeybee genetics in five main beekeeping regions with different climate. They reported colonies with mitotypes from lineages A, C and M; more than 50% of the colonies presented honeybees with mitotypes from the A lineage, mostly in subtropical and tropical regions. However, A1 was the most frequent mitotype in their study [17]. Likewise, Kraus et al. found in Veracruz, with tropical savanna climate in Mexico, honeybee mitotypes from the lineages A, C and M, but principally A1 and A4. In the lower part of the study site, samples were taken from hives established from locally captured swarms, indicating A mitotypes (67%) and European mitotypes (33%), meanwhile in the high altitudes (up to 2500 m) only feral colonies were sampled and the bee population was a 97% Africanized. They discussed these findings, that in the lower zones some beekeepers introduced constantly European queens, whereas in the higher altitudes Africanized bees replaced European bees, because of missing beekeeping activities [40].

The low diversity of mitotypes of the A lineage reported in this study could be explained by the action of the restriction enzyme *Dra*I that is known to miss genetic variations in the A lineage [8, 9, 43] and still there is one unidentified mitotype from the Northern Highlands.

The sequencing of the intergenic region of the COI-COII confirmed the results of identified mitotypes by digestion with the restriction enzyme *Dra*I. Both mitotypes A4 and A1 showed divergence between the samples from Nicaragua. Three bees in the present study were classified by sequencing as *A. mellifera scutellata* mitotype A4. The phylogenetic analysis indicated a similarity > 99.7% to sequences from Africa [39], Brazil [42] and Argentina [44], reinforcing the findings that this mitotype is found more frequently in subtropical and tropical areas [42].

Of the six mitotypes A1 from Nicaragua, five presented 100% similarity in between them, while one sample (MW695401) presented divergence (0.002). In this case, sequencing

comparison is useful to identify genetic variants within the same mitotype, which cannot be evidenced by digestion pattern of the *Dra*I test [39]. Remarkably, these five sequences were 100% similar to mitotype A1 from Argentina [45] and 4/5 to an A1 sequence from Benin, Africa [39]. Meanwhile the similarity to samples from south-western Europe and North Africa was lower (>99.6% and 99.4% respectively) [8, 46], not supporting that Nicaraguan A1 mitotypes may reflect Iberian honeybees introduced by the colonists [8]. In this discussion about the 'pre-Africanization' population, referring to mitotype A1 as *A. mellifera iberiensis* from the Iberian Peninsula, Kraus et al. described that in samples collected in Tamaulipas, Mexico, before the arrival of the Africanized honeybee in 1988, 13% of the honeybees presented mitotype A1 [40]. On the other hand, Chávez-Galarza et al. referred to "*Ancestral Haplotypes*" as mitotypes introduced from Africa to the Iberian Peninsula thousands of years ago, as part of the natural migration of *A. mellifera* [46]. Numerous studies about the subspecies from the Iberian Peninsula provide a complex dataset about phylogeographic patterns of *A. mellifera* [47], but still there are many questions about the origin of actual mitotypes from the A lineage detected in the Americas.

One characteristic for A1 mitotype belonging to *A. mellifera iberiensis* from south-western Europe could be the size of the forewing, commonly European subspecies have longer wings [48]. Conversely, in Nicaragua, colonies with mitotypes A4, generally characterized by smaller wings, here had even longer forewings than A1. These results cast doubt on the assumption that those bees are descendants from European honeybees, but only forewing length is not reliable to discriminate definitive subspecies, approaches like geometric morphometrics and if affordable combined to nuclear DNA markers, offer more information about the evolutionary processes [47].

Another important finding of this study disclosed, that sample MW695396 (mitotype A4) and sample MW695399 (mitotype A1) were collected from the same hive and presented a divergence of 0.119. In theory, all worker bees in a colony should have the same mtDNA, but beekeepers' practices, like interchanging frames from brood chambers, may contribute to find more than one mitotype per hive. That means, sampling only one worker bee per hive may be insufficient to estimate the diversity of mitotypes [27].

## Conclusion

In Nicaragua, regardless of the zone, there is a high degree of Africanization in *A. mellifera* colonies, represented by the mitotypes A1 and A4 and supported by the overall average of the right forewing length. The tropical climate prevailing throughout the country is a relevant factor for the habitat of Africanized bees, with mitotype A4 predominating at higher altitudes. At the same time, the colonies with the most defensive behaviour were reported in the Northern Highlands; defensiveness as a typical Africanized trait for honeybee colonies of the A4 lineage such as found in *A. mellifera scutellata* colonies. Beekeeping activities like queen replacements show a high impact in honeybee genetics, and this is probably why in Nicaragua, due to very rustic beekeeping, there has been a complete replacement of European bees by Africanized bees, at least no European lineages were detected by mtDNA analysis in the present study.

Nevertheless, Africanization in bee colonies presents advantages over European bees, related to resistance and/or tolerance to certain diseases mainly varroosis. These abilities are associated to a better hygienic behaviour and grooming, as well as a lower susceptibility to suffer invasion and reproduction of pathogens [9, 30, 31].

## Supporting information

**S1 Fig. Measurement of the right forewing with image tool 3.**
(TIFF)

**S1 Data. Data information, Africanized honeybee population in Nicaragua, forewing length and mitotype lineages.**
(XLSX)

## Acknowledgments

The authors thank to the beekeepers for their collaboration in the fieldwork and for the permission to investigate on their beehives.

## Author Contributions

**Conceptualization:** Christiane Düttmann, Byron Flores, Jessica Sheleby-Elías, Jorge Demedio.

**Data curation:** Christiane Düttmann, Byron Flores, Jessica Sheleby-Elías, Daymara Rodriguez, Matías Maggi, Jorge Demedio.

**Formal analysis:** Christiane Düttmann, Byron Flores, Gladys Castillo, Daymara Rodriguez, Matías Maggi.

**Investigation:** Christiane Düttmann, Byron Flores, Daymara Rodriguez, Jorge Demedio.

**Methodology:** Christiane Düttmann, Byron Flores, Jessica Sheleby-Elías, Gladys Castillo, Matías Maggi, Jorge Demedio.

**Supervision:** Christiane Düttmann, Byron Flores, Matías Maggi.

**Validation:** Christiane Düttmann.

**Visualization:** Christiane Düttmann, Byron Flores.

**Writing – original draft:** Christiane Düttmann, Byron Flores, Jessica Sheleby-Elías, Gladys Castillo, Daymara Rodriguez, Matías Maggi, Jorge Demedio.

**Writing – review & editing:** Christiane Düttmann, Byron Flores, Jessica Sheleby-Elías, Gladys Castillo, Daymara Rodriguez, Matías Maggi.

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
