## [Decision Letter · Decision Letter 0]

10 Dec 2021

PONE-D-21-31700Africanized honeybee population (Apis mellifera L.) in Nicaragua: Forewing length and mitotype lineages.PLOS ONE

Dear Dr. Flores,

Thank you for submitting your manuscript to PLOS ONE. After careful consideration, we feel that it has merit but does not fully meet PLOS ONE’s publication criteria as it currently stands. Therefore, we invite you to submit a revised version of the manuscript that addresses the points raised during the review process.

We look forward to receiving your revised manuscript.

Kind regards,

Tiago M. Francoy, Ph.D.

Academic Editor

PLOS ONE

Journal Requirements:

3. We note that Figure 2 in your submission contain copyrighted images. All PLOS content is published under the Creative Commons Attribution License (CC BY 4.0), which means that the manuscript, images, and Supporting Information files will be freely available online, and any third party is permitted to access, download, copy, distribute, and use these materials in any way, even commercially, with proper attribution. For more information, see our copyright guidelines: http://journals.plos.org/plosone/s/licenses-and-copyright.

4. We note that Figure 5 in your submission contain [map/satellite] images which may be copyrighted. All PLOS content is published under the Creative Commons Attribution License (CC BY 4.0), which means that the manuscript, images, and Supporting Information files will be freely available online, and any third party is permitted to access, download, copy, distribute, and use these materials in any way, even commercially, with proper attribution. For these reasons, we cannot publish previously copyrighted maps or satellite images created using proprietary data, such as Google software (Google Maps, Street View, and Earth). For more information, see our copyright guidelines: http://journals.plos.org/plosone/s/licenses-and-copyright.

a. You may seek permission from the original copyright holder of Figure 5 to publish the content specifically under the CC BY 4.0 license.  

Reviewers' comments:

Reviewer's Responses to Questions

**Comments to the Author**

1. Is the manuscript technically sound, and do the data support the conclusions?

Reviewer #1: Yes

Reviewer #2: Yes

2. Has the statistical analysis been performed appropriately and rigorously? 

Reviewer #1: Yes

Reviewer #2: Yes

3. Have the authors made all data underlying the findings in their manuscript fully available?

Reviewer #1: Yes

Reviewer #2: No

4. Is the manuscript presented in an intelligible fashion and written in standard English?

Reviewer #1: Yes

Reviewer #2: Yes

5. Review Comments to the Author

Reviewer #1: General comments:

The manuscript discusses the Africanization level of honeybee (Apis mellifera 1 L.) from Nicaragua using phenotypic, molecular, behavior, and morphometric analyses. I believe that this manuscript can be published in this Journal, but it is necessary to do a major reviewer. The authors need to organize better all methods that they used. They also need to work more in the discussion.

I suggest the authors add geometric morphometrics analysis to study shape and centroid size of the forewings because it can provide more accurate results than the traditional morphometrics that they used.

#Abstract

The authors need to work more on the abstract and include the question and the main objective of the study in the first paragraph.

Line 28: the authors used four methods to study honeybee populational variation. Please add the phenotypic and behavior in this line.

Line 28: Replace “approach” with approaches

# Introduction

Line 48: Replace “approach” with approaches

Line 51: replace “analysis” with analyses

Line 51-56: please extend this literature reviewer

Line 102: Choose “behaviour” (UK) or “behavior” (USA) in your manuscript.

Line 115-118: which previous studies did you use to elaborate your hypotheses? Please, include them.

#Methods

Line 133: are bees collected in the same year?

Lines 145-154: I suggest this paragraph be a subtitle about behavior

Lines 151-153: “Further information was recorded about the applied methods of multiplication of the beehives, queen bee replacement and the origin of the queen in the beehive”. What did you use to get this information?

Line 163: Are the “146 samples” the beehives? Please, specify it.

Line 166-172: Please provide an image of the forewing indicating the measurements. You also have two variables: colour and wing length. How many bees of the same colour did you use to measure the forewings? It is interesting to investigate if there are differences among bee forewings with different behaviors.

Line 172: Add the references of the software.

Line 176-182: why did you include samples in your work just for molecular analyses?

Lines 183-186: why did you include a new hypothesis here? Which previous studies did you use to elaborate this hypothesis?

Results

Line 263: you also studied the abdominal colour

Lines 274-283: the authors need to describe this procedure in the methods. How did you get this result?

Line 294: Write the species name in italic. Please, look at all manuscript.

Lines 300-306: present only your results and move this information to discussion.

Discussion

The authors need to discuss all result topics including the evolutionary history of Apis mellifera.

Lines 480-482: the authors agreed that geometric morphometrics can “offer more information about the evolutionary processes”. However, they used traditional morphometrics.

Reviewer #2: The process of the Africanization of Apis mellifera and its occupation in the Americas is a fascinating phenomenon that still needs to be studied. Information on the genetic and morphological diversity of the honeybee in Nicaragua contributes to a greater understanding of this process.

Here are some considerations:

- On line 55, it would be interesting to mention what these recent techniques are about. Unfortunately, in the text, it was not clear.

- Among a series of measures proposed by Ruttner in traditional morphometry to distinguish Apis mellifera lineages, do you think choosing only one linear measurement (affected by environmental factors) is enough to discriminate the groups? For example, according to Diniz-Filho et al., 2000 “Phylogenetically, the wing venation is more informative compared to the more environment–sensitive character categories of size, colour or pilosity”.

- In the text, the average length of the wing of European honeybees was not mentioned. Are they larger or smaller than the measurements of Africanized bees?

- Why was the entire sample of phenotypic variation not included in the molecular analyses? Despite the description of the criteria “...phenotypic diversity within the colony and mean length of the forewing.”, which factor determined this choice was unclear.

- On line 197, describe the modifications in the protocol.

- Neighbor-joining is a clustering algorithm that clusters haplotypes based on genetic distance. Maximum likelihood can apply a model of sequence evolution and is better for building a phylogeny using sequence data. Neighbor-joining isn’t a phylogenetic analysis

-Paragraphs 262-266 would fit better at the end of the text as a conclusion to the study.

6. PLOS authors have the option to publish the peer review history of their article (what does this mean?). If published, this will include your full peer review and any attached files.

Reviewer #1: No

Reviewer #2: No

---

## [Author Response · Author response to Decision Letter 0]

10 Jan 2022

León, 10th January 2022

Manuscript Number: PONE-D-21-31700

Dear Tiago M. Francoy, Ph.D.

Academic Editor

PLOS ONE

We are resubmitting our manuscript Africanized honeybee population (Apis mellifera L.) in Nicaragua: Forewing length and mitotype lineages. to PLOS ONE and all corrections are included. All the authors have reviewed the draft and approved its contents and conclusions.

Best regards,

The authors.

Dear Academic Editor, dear Reviewers.

Thank you for considering publishing our manuscript and for believing in the scientific merit of our contribution through this investigation. We have carefully revised your comments and suggestions and made adjustments accordingly to improve the manuscript. If you do not mind, we would first like to explain briefly the scientific situation of beekeeping in Nicaragua.

As a few academics in Veterinary Medicine from UNAN-León (National Autonomous University of León), we started studies on varroosis in 2004, following a personal request from some beekeepers. At the time, we were not beekeeping specialists and we had no previous references from published articles on beekeeping issues in Nicaragua. Therefore, we started to learn about these insects and their importance for us.

With the material and data from a cross-sectional study between 2013 and 2016, we had the first opportunity to contribute scientific information on beekeeping topics in Nicaragua. For the past 10 years, we worked together with specialists in Latin America to share our knowledge with the aim of improving our studies, above all to offer scientific solutions to practical problems in Nicaraguan beekeeping.

We have learned from “our bees” that they are very healthy and strong and that they have been able to defend themselves against diseases so far. (see article Düttmann C, Flores B, Sheleby-Elías J, Castillo G, Osejo H, Bermudez S, et al. Morphotype and haplotype identification of Varroa destructor (Acari: Varroidae), and its importance for apiculture in Nicaragua. Exp Appl Acarol. 2021;83: 527–544. doi:10.1007/s10493-021-00603-9).

In addition, we have learned from the Nicaraguan beekeepers that most of them practice rustic beekeeping, but as academic supervisors, we have good contact and acceptance with our observations on improving management and honey production. The process of Africanization in Nicaragua is of global concern. Rustic beekeeping mainly leaves it to nature to influence the population genetics of Apis mellifera. We therefore expect an almost “pure” Africanized bee population.

Your principal question why we didn´t applied geometric morphometrics is that in the first place we wanted to realize an estimation about the Africanization level in the honeybee population of Nicaragua, expecting a high degree. Therefor the method of Rinderer is the less time and money consuming one with valid results. On the other hand, we wanted to confirm these preliminary results with genetic analysis, starting to get an insight in population genetics in Nicaraguan honeybee colonies. Our limitations are due to our financial situation; however, we look forward to realize further research with approaches that are more detailed.

We are pleased to say that we have effected all necessary changes and corrections to your comments as you see in the following part of the responses to the reviewers and in the attached document named ‘Revised Manuscript with Track Changes’. As advised, we have also attached a separate document named ‘Manuscript’ that do not have track changes. Please do not hesitate to get back to us if there is any need to do so.

Best regards

The authors

 

Editor and Reviewer comments:

Journal Requirements:

Resp: Corrected. We ensured that our manuscript meets PLOS ONE's style requirements, including those for file naming.

Resp: Corrected. We provided all data and additional material as files of the supporting information (file name: “S1_Data.XLSX” and “S1_Fig.TIFF”).

3. We note that Figure 2 in your submission contain copyrighted images. All PLOS content is published under the Creative Commons Attribution License (CC BY 4.0), which means that the manuscript, images, and Supporting Information files will be freely available online, and any third party is permitted to access, download, copy, distribute, and use these materials in any way, even commercially, with proper attribution. 

Resp: Corrected. Dear editor, the photograph in figure 2 is not copyrighted, we took photographs of the samples in the laboratory, when we started the diagnostic procedures. When the article is published, we agree that our material can be used under Creative Commons Attribution License (CC BY 4.0).

This is the reference for figure 2 (Fig2.TIF):

Düttmann, C. (Photographer). 2013, November, León, Nicaragua. Honeybees sampled from one beehive in Boaco, sample no. A8-04-10-13. Laboratory CEVEDI – UNAN - León.

In the same way the new figure we provide for Supporting Information, as suggested by reviewer #1, is another photograph that is not copyrighted. We made the photograph using a stereoscope to determine the forewing length of the sampled honeybees and now we edited it to indicate measurement practice.

This is the reference for figure S1 (S1_Fig.TIFF):

Düttmann, C. (Photographer). 2016, September, León, Nicaragua. Right forewing of Apis mellifera detached for length measurement, Laboratory CEVEDI – UNAN - León.

4. We note that Figure 5 in your submission contain [map/satellite] images which may be copyrighted. All PLOS content is published under the Creative Commons Attribution License (CC BY 4.0), which means that the manuscript, images, and Supporting Information files will be freely available online, and any third party is permitted to access, download, copy, distribute, and use these materials in any way, even commercially, with proper attribution. For these reasons, we cannot publish previously copyrighted maps or satellite images created using proprietary data, such as Google software (Google Maps, Street View, and Earth). For more information, see our copyright guidelines: http://journals.plos.org/plosone/s/licenses-and-copyright.

Resp: Corrected. We replaced the copyrighted map in figure 5 with a map from the following source: naturalearthdata.com. "All versions of Natural Earth raster + vector map data found on this website are in the public domain. You may use the maps in any manner, including modifying the content and design, electronic dissemination, and offset printing. The primary authors, Tom Patterson and Nathaniel Vaughn Kelso, and all other contributors renounce all financial claim to the maps and invites you to use them for personal, educational, and commercial purposes. No permission is needed to use Natural Earth. Crediting the authors is unnecessary.”

Comments to the Author:

1. Is the manuscript technically sound, and do the data support the conclusions?

Reviewer #1: Yes

Reviewer #2: Yes

2. Has the statistical analysis been performed appropriately and rigorously?

Reviewer #1: Yes

Reviewer #2: Yes

3. Have the authors made all data underlying the findings in their manuscript fully available?

Reviewer #1: Yes

Reviewer #2: No

4. Is the manuscript presented in an intelligible fashion and written in standard English?

Reviewer #1: Yes

Reviewer #2: Yes 

5. Review Comments to the Author

Reviewer #1: 

#General comments:

The manuscript discusses the Africanization level of honeybee (Apis mellifera 1 L.) from Nicaragua using phenotypic, molecular, behavior, and morphometric analyses. I believe that this manuscript can be published in this Journal, but it is necessary to do a major reviewer. The authors need to organize better all methods that they used. They also need to work more in the discussion.

Resp: Dear reviewer. Thank you for your consideration and your constructive suggestions. As you will see in the following points, we added the lacking information in method, we improved the reporting of the results and we enriched the discussion.

I suggest the authors add geometric morphometrics analysis to study shape and centroid size of the forewings because it can provide more accurate results than the traditional morphometrics that they used.

Resp: Dear reviewer, thank you for this suggestion. Of course, the study would have shown more results if we had carried out a geometric-morphometric analysis, but the main aim of the study was not to define subspecies by morphometric features, but to determine the degree of Africanization of the honeybees in Nicaragua by the length of right forewing as described by Rinderer et al. 1986. Phylogenetic lineages were determined by mtDNA analysis to confirm the results and to provide initial insights into honeybee populations.

#Abstract

Include the question and the main objective of the study in the first paragraph.

Resp: Corrected. We included the principle question and the main objective of the study in the first paragraph of the abstract.

Line 28: the authors used four methods to study honeybee populational variation. Please add the phenotypic and behavior in this line.

Resp: Corrected. We added that we related beehive characteristics and management to the results of morphometric and genetic analyses. 

Line 28: Replace “approach” with approaches.

Resp: Corrected. We revised the manuscript and we corrected approach to approaches where it was necessary. 

# Introduction

Line 51: replace “analysis” with analyses.

Resp: Corrected. We revised the manuscript and we corrected analysis to analyses where it was necessary.

Line 51-56: please extend this literature reviewer.

Resp: Corrected. We have extended our statement about the various morphometric and genetic approaches for the determination of Apis mellifera species, subspecies and populations with brief explanations and added further references to support them.

Line 102: Choose “behaviour” (UK) or “behavior” (USA) in your manuscript.

Resp: Corrected.

Line 115-118: which previous studies did you use to elaborate your hypotheses? Please, include them.

Resp: Corrected. We cited three studies from Costa Rica, Mexico and USA where the factors are discussed that support our hypothesis.

#Methods

Line 133: are bees collected in the same year?

Resp: Corrected. In the text, we included the period sampling was conducted, but the years are not relevant for statistical analysis.

Lines 145-154: I suggest this paragraph be a subtitle about behavior

Resp: Corrected. Subtitle added “Data collection about defensive behaviour and beehive management”

Lines 151-153: “Further information was recorded about the applied methods of multiplication of the beehives, queen bee replacement and the origin of the queen in the beehive”. What did you use to get this information?

Resp: Corrected. During the sampling, additional data were recollected using a questionnaire applied to the beekeepers who manage the selected apiary. The questionnaire was previously pilot-tested.

Line 163: Are the “146 samples” the beehives? Please, specify it.

Resp: Corrected. We improved the sentence for better understanding:

As the first feature in this study, phenotypic diversity of the bee colonies in each of the 146 hives was examined using photographic identification of the collected bees to document uniformity or diversity of the abdominal colour pattern.

Line 166-172: Please provide an image of the forewing indicating the measurements. You also have two variables: colour and wing length. How many bees of the same colour did you use to measure the forewings? It is interesting to investigate if there are differences among bee forewings with different behaviors.

Resp: Corrected. A) We added an image of the forewing measurement and included it to Supporting Information (S1_Fig.TIFF).

B) As Ruttner (1988) described, differences in colour among bee colonies is a bimodal variation, therefore our selection criteria to measure the forewing length, was based on the phenotypic diversity in the sample, including the same number of bees from each colour. We added information that is more detailed in methods.

C) We studied the association between the result of the mean forewing length of a bee colony and the beehive characteristics such as the origin of the hive and the defensive behaviour, as well as beekeeping practices. “There was no significant difference between the characteristics of the beehives and management activities according to the forewing length (ANOVA, origin of the colony: F3,146 = 0.13, p > 0.05; abdominal colour pattern: F1,146 = 0.55, p > 0.05; defensiveness of the beehive: F2,146 = 1.52, p > 0.05; queen replacement: F 1,146 = 0.48, p > 0.05; origin of the queen: F2,146 = 0.66, p > 0.05) (Fig 4)”

Line 172: Add the references of the software.

Resp: Corrected.

Line 176-182: why did you include samples in your work just for molecular analyses?

Resp: Dear reviewer, we included to our own sample collection during the study, samples from five different sentinel apiaries, which are of interest because of their geographic position “northern border to Honduras (1), southern border to Costa Rica (1), Island in the Nicaraguan Lake (1) and the South Caribbean Coast (2)”. These five locations could have been a possible provenience for different genetic findings. The samples were obtained directly from the Institute of Agricultural Protection and Health (IPSA), but included only few specimen for diagnostic procedure, therefore additional information was not available, such as phenotypic characters or forewing length measurement and on the other hand the lack of data from the questionnaire. However, the genetic data is important to get an insight into the existing lineages of Apis mellifera in Nicaragua.

Lines 183-186: why did you include a new hypothesis here? Which previous studies did you use to elaborate this hypothesis?

Resp: Dear reviewer, this is not a new hypothesis, here we underline our assumption that exchanging frames with brood between beehives to strengthen the weak hives could be a reason to find different mitotypes in the same beehive.

As Evans et al. indicated in “Standard methods for molecular research in Apis mellifera”, that “Due to the risk of drifting between colonies” (even if they mention natural drifting) “it is arguably worth sampling more than one individual to avoid mistakes in assigning colony heritage.” The impact of hive management on genetic pattern is still very important to investigate and the exchanging of brood combs is a widespread practice in Nicaragua.

#Results

Line 263: you also studied the abdominal colour

Resp: Dear reviewer, this is correct and we pointed it out in line 266.

Lines 274-283: the authors need to describe this procedure in the methods. How did you get this result?

Resp: Corrected. We added in methods “Management information was recorded from the beekeepers about the applied methods of multiplication of the beehives, queen bee replacement and the origin of the queen in the beehive. To collect data, the investigators developed a questionnaire that was pilot – tested before application.”

Line 294: Write the species name in italic. Please, look at all manuscript.

Resp: Corrected.

Lines 300-306: present only your results and move this information to discussion.

Resp: Corrected.

# Discussion

The authors need to discuss all result topics including the evolutionary history of Apis mellifera.

Resp: Dear reviewer, thank you for your comment. In the discussion section, we added a paragraph discussing the characteristics of beehives and the management of apiaries. In regards to the evolutionary history, we believe we have addressed all aspects relevant to our results in the discussion.

Lines 480-482: the authors agreed that geometric morphometrics can “offer more information about the evolutionary processes”. However, they used traditional morphometrics.

Resp: Dear reviewer, here we mentioned that for subspecies discrimination, it is necessary to extent the approach to geometric morphometrics, but our aim was to distinguish Africanized from European honeybee colonies. In this context (line 480-482), the discussion was about the longer forewings we found in bees with an A4 mitotype compared to those bees with A1 mitotype. If we consider that, the A1 mitotype that we sent to GenBank was identified as A. mellifera iberiensis and if we want to relate this finding to a morphometric approach, it would be necessary to apply geometric morphometrics to discriminate subspecies. Nevertheless, this was not the aim of our study; however, it is a relevant indication for future studies.

Reviewer #2: 

The process of the Africanization of Apis mellifera and its occupation in the Americas is a fascinating phenomenon that still needs to be studied. Information on the genetic and morphological diversity of the honeybee in Nicaragua contributes to a greater understanding of this process.

Resp: Dear reviewer. Thank you for your comment; we share the same enthusiasm about Apis mellifera research and we want to fill an information gap in Central America with our first study.

Line 55: it would be interesting to mention what these recent techniques are about. Unfortunately, in the text, it was not clear.

Resp: Corrected. The same observation made reviewer #1. As you can see in our response to him/her, we have extended our statement about the various morphometric and genetic approaches for the determination of Apis mellifera species, subspecies and populations with brief explanations and added further references to support them.

Line 197: describe the modifications in the protocol.

Resp: Corrected. We added that we changed in the elution volume of the samples (from 200 µl to 50 µl for each sample), a common practice to decrease elution volume in genetic approaches. We decided to do this modification, because in our experience we noted better results with this elution volume. Decreasing elution volumes does not significantly compromise DNA yields (El-Mogy et al. The Effect of Elution Volume on DNA Quantity and Quality Using Norgen’s Saliva DNA Isolation Kit. 2016).

Lines 262-266: would fit better at the end of the text as a conclusion to the study.

Resp: Dear reviewer, thank you for this suggestion, but we regard these 5 lines as an introduction to the following results, a kind of summary which results are to be expected. However, we improved the conclusions concerning your observation.

General observations:

- Among a series of measures proposed by Ruttner in traditional morphometry to distinguish Apis mellifera lineages, do you think choosing only one linear measurement (affected by environmental factors) is enough to discriminate the groups? For example, according to Diniz-Filho et al., 2000 “Phylogenetically, the wing venation is more informative compared to the more environment–sensitive character categories of size, colour or pilosity”.

Resp: Dear reviewer, As we mentioned in our manuscript, now Line 112 - 114: “Even assuming that this technique (note: measurement of forewing length) provides only preliminary results [13] and that it is unsuitable for identifying subspecies, it does indicate the probability that a colony is Africanized [21].” 

As we mentioned in our initial letter and in the response to reviewer #1, the main aim of our study was to identify the degree of Africanization in Nicaragua and the mitotype lineages distributed in the different zones where apiculture is most frequent.

Ruttner established an extended Morphometric Bee Data Bank in Oberursel, Germany (the Ruttner collection with international participation) to discriminate groups, we are far from this, but with our study we can contribute to future research hypothesis. Another factor is, that due to our financial and technical limitations, we chose the simplified technique described by Rinderer et al., using a single morphometric character to differentiate just between Africanized and European bee colonies. The principal result of the study Rinderer et al. “Field and Simplified techniques for identifying Africanized and European honeybees, 1986”, showed that the forewing length was “the best single character to distinguish between Africanized and European honeybees: 86% were correctly identified and no sample was misidentified; the remaining were unidentified.” Based on the morphometric results, we provided data that are more accurate for identifying lineages by mtDNA analysis and sequencing.

- In the text, the average length of the wing of European honeybees was not mentioned. Are they larger or smaller than the measurements of Africanized bees?

Resp: Corrected. We added that the overall average of the forewing length in European honeybees is � = 9.20 mm, larger than the Africanized bees. Reference: 

Rinderer TE, Sylvester HA, Brown MA, Villa JD, Pesante D, Collins AM, et al. Field and simplified techniques for identifying africanized and european honey bees. Apidologie. 1986;17: 33–48. doi:10.1051/apido:19860104

- Why was the entire sample of phenotypic variation not included in the molecular analyses?

Resp: Dear reviewer, all the samples with complete information were included and analysed, inclusive the phenotypic variation. However, additionally we included 14 honeybees from five sentinel apiaries provided by IPSA, the Institute of Agricultural Protection and Health (see answer to line 176 – 182 reviewer #1).

Despite the description of the criteria “...phenotypic diversity within the colony and mean length of the forewing.”, which factor determined this choice was unclear.

Resp: Corrected. After a carefully revision, we saw were the problem of understanding of our sample size for molecular diagnosis is. We changed the order of the paragraphs and we explained the procedure better.

For molecular diagnosis, honeybees from the saved 146 samples were selected from all inspected apiaries in study. The criteria heterogeneity of the abdominal colour within the colony and mean length of forewing were indicative for selection. However, first we had to mention that we decided to make PCR diagnosis with two worker bees instead of one (see Evans et al. Standard methods for molecular research in Apis mellifera). In our previous submitted manuscript, we explained this part in the second paragraph, so this led to confusion. The new redaction avoid this misunderstanding.

- Neighbor-joining is a clustering algorithm that clusters haplotypes based on genetic distance. Maximum likelihood can apply a model of sequence evolution and is better for building a phylogeny using sequence data. Neighbor-joining isn’t a phylogenetic analysis

Resp: Corrected. Thank you for this suggestion; we revised literature and we agree that ML method is accurate to determine evolutionary divergences. We substituted Neighbor-joining by the Maximum likelihood method and we updated section methods as well as the results for the phylogenetic tree.

---

## [Editor Report · Decision Letter 1]

12 Apr 2022

Africanized honeybee population (Apis mellifera L.) in Nicaragua: Forewing length and mitotype lineages.

PONE-D-21-31700R1

Dear Dr. Flores,

We’re pleased to inform you that your manuscript has been judged scientifically suitable for publication and will be formally accepted for publication once it meets all outstanding technical requirements.

Kind regards,

Tiago M. Francoy, Ph.D.

Academic Editor

PLOS ONE

Additional Editor Comments (optional):

After carefully reading the new version of the manuscript and the anwers provided by the authors, I think all the requirements asked by the reviewers are satisfactory. It is my opinion that the manuscript is ready for acceptance and publication.
---

## [Editor Report · Acceptance letter]

14 Apr 2022

PONE-D-21-31700R1 

Africanized honeybee population (*Apis mellifera* L.) in Nicaragua: Forewing length and mitotype lineages. 

Dear Dr. Flores:

I'm pleased to inform you that your manuscript has been deemed suitable for publication in PLOS ONE. Congratulations! Your manuscript is now with our production department. 

Kind regards, 

on behalf of

Dr. Tiago M. Francoy 

Academic Editor

PLOS ONE